# Side chain to main chain hydrogen bonds stabilize a polyglutamine helix in a transcription factor

Albert Escobedo[1,2,10], Busra Topal[1,2,10], Micha B.A. Kunze[3], Juan Aranda [1,2], Giulio Chiesa[1,2], Daniele Mungianu[1,2], Ganeko Bernardo-Seisdedos [4], Bahareh Eftekharzadeh[1,2], Margarida Gairí[5], Roberta Pierattelli [6], Isabella C. Felli [6], Tammo Diercks[4], Oscar Millet[4], Jesús García[1], Modesto Orozco[1,2,7], Ramon Crehuet [8], Kresten Lindorff-Larsen [3] & Xavier Salvatella [1,2,9]

Polyglutamine (polyQ) tracts are regions of low sequence complexity frequently found in transcription factors. Tract length often correlates with transcriptional activity and expansion beyond specific thresholds in certain human proteins is the cause of polyQ disorders. To study the structural basis of the association between tract length, transcriptional activity and disease, we addressed how the conformation of the polyQ tract of the androgen receptor, associated with spinobulbar muscular atrophy (SBMA), depends on its length. Here we report that this sequence folds into a helical structure stabilized by unconventional hydrogen bonds between glutamine side chains and main chain carbonyl groups, and that its helicity directly correlates with tract length. These unusual hydrogen bonds are bifurcate with the conventional hydrogen bonds stabilizing α-helices. Our findings suggest a plausible rationale for the association between polyQ tract length and androgen receptor transcriptional activity and have implications for establishing the mechanistic basis of SBMA.

[1] Institute for Research in Biomedicine (IRB Barcelona), The Barcelona Institute of Science and Technology, Baldiri Reixac 10, 08028 Barcelona, Spain. [2] Joint BSC-IRB Research Programme in Computational Biology, Baldiri Reixac 10, 08028 Barcelona, Spain. [3] Structural Biology and NMR Laboratory, Linderstrøm-Lang Centre for Protein Science, Department of Biology, University of Copenhagen, 2200 Copenhagen, Denmark. [4] CIC bioGUNE, Bizkaia Science and Technology Park bld 801A, 48160 Derio, Bizkaia, Spain. [5] NMR Facility, Scientific and Technological Centers University of Barcelona (CCiTUB), Baldiri Reixac 10, 08028 Barcelona, Spain. [6] CERM and Department of Chemistry "Ugo Schiff", University of Florence, Via Luigi Sacconi 6, Sesto Fiorentino, 50019 Florence, Italy. [7] Department of Biochemistry and Biomedicine, University of Barcelona, Avinguda Diagonal 645, 08028 Barcelona, Spain. [8] Institute for Advanced Chemistry of Catalonia (IQAC-CSIC), Jordi Girona 18-26, 08034 Barcelona, Spain. [9] ICREA, Passeig Lluís Companys 23, 08010 Barcelona, Spain. [11] These authors contributed equally: Albert Escobedo, Busra Topal. Correspondence and requests for materials should be addressed to R.C. (email: ramon.crehuet@iqac.csic.es) or to K.L.-L. (email: lindorff@bio.ku.dk) or to X.S. (email: xavier.salvatella@irbbarcelona.org)

Polyglutamine (polyQ) tracts are low-complexity regions that are composed almost exclusively of Gln residues. They are frequent in the human proteome, particularly in the intrinsically disordered domains of proteins involved in the regulation of transcription, such as the activation domains of transcription factors[1]. The biological function of polyQ tracts is not well understood, but it has been suggested that they regulate the activity of the proteins that harbor them by modulating the stability of the complexes that they form[2]. The lengths of polyQ tracts are variable because their coding DNA sequences tend to adopt secondary structures that hamper replication and repair[3]. Contractions and expansions in polyQ tracts can have functional consequences, and the lengths of the tracts may have been subject to natural selection[4]. As an example, it has been proposed that the length of the polyQ tract present in the protein huntingtin correlates with the intellectual coefficient[5], presumably because it plays important although still not well-defined roles in neural plasticity[6].

For nine specific proteins, the variability in the lengths of polyQ tracts has pathogenic implications. Expansions beyond protein-specific thresholds are associated with nine rare hereditary neurodegenerative diseases known as polyQ diseases[7]. The mechanistic basis of this phenomenon is a matter of debate. Some authors have proposed that the expanded transcripts themselves are the neurotoxic species[8] due, possibly, to their propensity to phase separate[9], while others have suggested that expanded polyQ proteins are inherently neurotoxic[10]. It is widely believed, however, that polyQ expansions cause neurotoxicity because they decrease protein solubility, which in turn leads to the formation of cytosolic or nuclear aggregates that interfere with proteasomal protein degradation[11] and sequester the transcriptional machinery[12]. This notion is supported by experiments carried out in vitro and in cells, which showed that polyQ expansion decreases protein solubility[13] and causes cell death[14], as well as in vivo, which revealed that enhanced polyQ aggregate clearance leads to improvements in the phenotypes induced by the polyQ expansion[15].

It has been hypothesized that polyQ tracts have the generic propensity to undergo a tract length-dependent conformational change producing a highly insoluble structure when their lengths reach the corresponding thresholds. A substantial number of theoretical, computational, and experimental studies have thus examined how the conformational properties of polyQ tracts change with their length. Some of these studies have suggested that expansions of the polyQ tract of huntingtin, associated with Huntington disease[16]—the most common polyQ disease—confer the ability to adopt extended conformations with β secondary structure[17]. By contrast, most experimental studies carried out to date report that polyQ tracts are collapsed disordered coils that barely change conformation upon expansion[18]. These observations led to an alternative hypothesis that proposes that expansion leads to toxicity by increasing the affinity of polyQ tracts for their interactors, regardless of conformation[19].

The androgen receptor (AR, 919 residues) is the nuclear receptor that regulates the development of the male phenotype. It harbors a polyQ tract whose helical propensity—that we have recently revealed by nuclear magnetic resonance (NMR) and circular dichroism (CD)[20]—increases upon expansion[20,21]. This tract is associated with the neuromuscular disease spinobulbar muscular atrophy (SBMA)[22], a condition that affects men with AR genetic variants coding for tracts with more than 37 residues, which form fibrillar cytotoxic aggregates[23]. The length of this tract also anti-correlates with the risk of suffering prostate cancer[24] due to its influence on AR transcriptional activity[25]. It therefore appears that the length of the polyQ tract of AR must be in a specific range in order to prevent the over-activation of the

receptor in prostate cancer and simultaneously minimize its propensity to form cytotoxic aggregates in SBMA. This trade-off is reflected in the distribution of AR polyQ tract lengths in the population, despite some variations between ethnic groups[26].

Although these sequence–activity relationships are relevant for understanding the causes of two diseases they have not been rationalized, in part due to the difficulty in obtaining atomic resolution structures of these poorly soluble repetitive sequences. By establishing robust assays and analysis procedures, we have generated conformational ensembles representing the structural properties of the polyQ tract of the AR[20] as a function of tract length by reweighting trajectories obtained by molecular dynamics (MD) on the basis of backbone chemical shifts measured by solution NMR spectroscopy, which we have validated by using CD spectroscopy and QM/MM (quantum mechanics/molecular mechanics) calculations. We find that the helicity of the tract correlates with its length as a result of the accumulation of unconventional interactions where Gln side chains donate a hydrogen bond to the main chain carbonyl of the residue at relative position $i-4$. In addition, we have found that the strength of these interactions depends on the ability of the side chain of the acceptor residue to shield the interacting moieties from competition with water molecules, indicating that the secondary structure of polyQ tracts can depend on their environment. Taken together, our results suggest explanations for how changes in polyQ tract length alter the transcriptional activity of AR and its propensity to aggregate in SBMA, thus providing plausible rationales for the range of tract lengths observed in men.

## Results

**The polyQ helix of AR gains stability upon expansion.** To confirm that the helical nature of the polyQ tract of AR[20] stems from local interactions and study how the N-terminal flanking region influences helicity, we analyzed the secondary structure of synthetic peptides $uQ_{25}$, $uL_4Q_{25}$, and $L_4Q_{25}$ (Fig. 1a) at pH 7.4 and 277 K by CD. Peptide $uQ_{25}$, where the letter u stands for uncapped, represents a polyQ tract containing 25 residues flanked by Lys residues, used to enhance solubility at physiological pH. $uL_4Q_{25}$ has four Leu residues found N terminally to the polyQ region in AR, while $L_4Q_{25}$ contains four additional AR residues (Pro-Gly-Ala-Ser), predicted to act as N-capping motif[27] (Supplementary Fig. 1). As shown in Supplementary Fig. 2, the spectra of $uL_4Q_{25}$ and in particular of $L_4Q_{25}$ have well-defined minima at ca. 205–208 and 222 nm, thereby indicating that they are 40 and 55% helical, respectively. These results show that the helicity of this polyQ tract stems from interactions involving eight residues flanking it at the N terminus, including a predicted N-capping motif and four Leu residues[20].

To quantify the extent to which helicity is determined by tract length, we used CD at pH 7.4 and 277 K to study polyQ peptides equivalent to $L_4Q_{25}$, but with tract lengths of 4, 8, 12, 16, and 20 residues ($L_4Q_n$, Fig. 1a) and observed that they are strongly correlated (Fig. 1b). Helicity increased abruptly from $L_4Q_4$ to $L_4Q_8$ and $L_4Q_{12}$, from ca. 5% to ca. 40%, and then increased slightly upon further expansion. Since the CD signal depends both on the amount and length of helical structures, and to determine the residue-specific distribution of helicity, we measured the backbone chemical shifts of the peptides by solution NMR at physiological pH and 278 K (Supplementary Figs. 3 and 4) and analyzed them with the algorithm δ2D[28]. In agreement with the results obtained by CD, helical propensity increased upon polyQ tract expansion (Fig. 1c), concomitant with a change in the identity of the residue with the highest helicity: whereas for $L_4Q_4$ this is L3, it shifts to L4 for $L_4Q_8$ and $L_4Q_{12}$ and to Q1 for $L_4Q_{16}$ and $L_4Q_{20}$ (Fig. 1c). Although it is likely that the

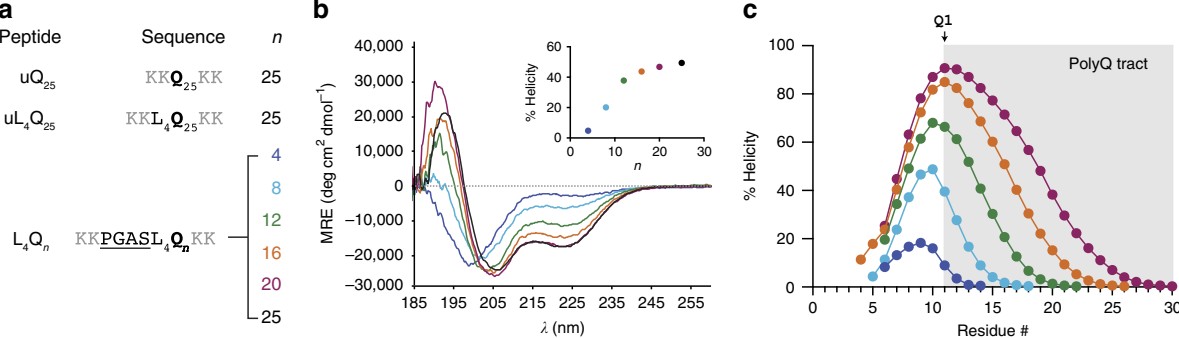

**Fig. 1** The stability of the androgen receptor (AR) polyQ helix increases upon tract expansion. **a** Sequences of the $uQ_{25}$, $uL_4Q_{25}$, and $L_4Q_n$ peptides used in this work. **b** Circular dichroism (CD) spectra of peptides $L_4Q_4$ to $L_4Q_{25}$ measured at pH 7.4 and 277 K and plot of the helicity determined by CD as a function of the size of the polyQ tract length, $n$ (inset, color coded). **c** Residue-specific helicity of peptides $L_4Q_4$ to $L_4Q_{20}$ obtained from an analysis of the backbone chemical shifts measured by nuclear magnetic resonance (NMR) at pH 7.4 and 278 K by using the algorithm δ2D[28] with an indication of the region of sequence corresponding to the polyQ tract and of its first residue

large differences in helicity that we observe will be attenuated at physiological temperature, we conclude that the helical propensity of this polyQ tract depends on its length and that the residue with the highest helicity can be part of the tract.

**The Gln side chains have a well-defined conformation.** To rationalize the stability of the polyQ helix, we extended our NMR analysis to the side chains and initially focused on the carboxamide groups of the Gln residues (Supplementary Fig. 5). We found that the $^{15}N$ side chain resonances of the homopolymeric polyQ sequences are well dispersed and that the associated chemical shifts correlate with their position in the sequence, that is, that the resonances of the first residue of the tract appear up-field (111.75 ppm for Q1 in $L_4Q_{20}$) (Fig. 2a) and shift to lower fields towards its C terminus (113.15 ppm for Q20 in $L_4Q_{20}$). Remarkably, the first four residues (Q1–Q4) have chemical shifts that are markedly lower; for example, in $L_4Q_{20}$ the difference in side chain $^{15}N$ chemical shift between Q4 and Q5 is 0.22 ppm, whereas the resonances of Q5 and Q6 overlap. These observations indicate that the chemical environment of the Gln side chains varies along the polyQ tract, especially for the first residues.

We then analyzed the $^1H$ resonances of the Gln side chains. Particularly in the first residues of the tract, the resonances of the γ protons, adjacent to the carboxamide group (Fig. 2b), overlap in the peptide with the shortest tract, but gradually split as the length of the tract increases to 20. The behavior of the β protons, which are instead adjacent to the peptide backbone (Fig. 2c and Supplementary Fig. 6), is more complex. In $L_4Q_4$ they are split, and upon tract expansion to $L_4Q_{12}$, they collapse into one peak, but then split again in $L_4Q_{16}$ and especially in $L_4Q_{20}$. These effects, caused by redistributions of side chain rotameric states, correlate with the increases in helicity that occur upon tract expansion reported in Fig. 1c and indicate that the conformations of the main chain and side chain of these residues are not independent. Although these effects are particularly marked for the first three or four residues of the tract, they can also be seen in the residues following them in the sequence, particularly in $L_4Q_{16}$ and $L_4Q_{20}$ (Supplementary Fig. 6). This indicates that, in a given peptide, the population of the side chain conformation causing the effects gradually decreases along the sequence. In summary, we found that the conformational properties of the side chains of Gln residues correlate with their helicity.

**Hydrogen bonds between Gln side chains and the main chain.** To rationalize these observations, we produced conformational ensembles for each peptide by using the NMR backbone chemical

shifts to reweight all atom MD trajectories. First, given that all peptides have fractional helicity, we sampled the conformational space available to them by generating fully helical conformations and simulating their dynamics for up to 5 μs at 300 K in explicit solvent. We observed that the helical starting structures had a lifetime that depended on the size of the system and that partially helical conformations were re-populated after unfolding even for the largest one, $L_4Q_{20}$ (Supplementary Fig. 7). These findings indicate that both the helical and unfolded states of the peptides were sampled under these conditions, which is necessary for reweighting procedures to perform well[29]. Then, we used the Cα and CO backbone chemical shifts to reweight the trajectories with a Bayesian/maximum entropy (BME) algorithm[30]. In this procedure, the degree of reweighting and, therefore, the extent to which the back-calculated chemical shifts agree with those measured experimentally is controlled by the parameter θ, which determines the balance between the prior information encoded in the MD trajectory and the experimental data (Supplementary Figs. 8 and 9 and, for $θ = 4$, Fig. 3b). We analyzed the secondary structure of the reweighted trajectories and found that their overall helicity increased with the length of the polyQ tract (Fig. 3a), as observed by CD (Fig. 1b). Furthermore, after reweighting the effect of expansion on the helicity of the various residues of the peptide was found to be equivalent to that observed by NMR, thereby indicating that the procedure yielded structural ensembles that are useful models of the conformational properties of polyQ peptides under the conditions used for the CD and NMR experiments (Fig. 3c).

The $^{15}N$ chemical shifts of backbone amides depend on the hydrogen bonding status of both the HN group and the adjacent CO[31]. We thus hypothesized that the high dispersion of $^{15}N$ Gln side chain resonances (Fig. 2a) is due to hydrogen bonding interactions of the carboxamide group of the Gln side chains. The primary amide ($NH_2$) groups of Gln and Asn side chains are good donors[32], and in folded proteins Gln residues have been found to donate hydrogens to the backbone COs preceding them in the sequence[33]. To study this possibility, we analyzed the hydrogen bonds formed by Gln side chains in the reweighted trajectories. We found that in the most frequent hydrogen bond the side chain of a Gln residue donates a hydrogen to the main chain CO group of the residue at relative position $i − 4$ in the sequence (Fig. 4a). This specific interaction, which we call $i → i − 4$ side chain to main chain hydrogen bond—that is, sc($i$) → mc($i − 4$), has been reported in protein structures deposited in the protein data bank (PDB). Interestingly, it occurs almost exclusively in α-helices both in the PDB[33] and in the trajectories (Supplementary Fig. 10), thus

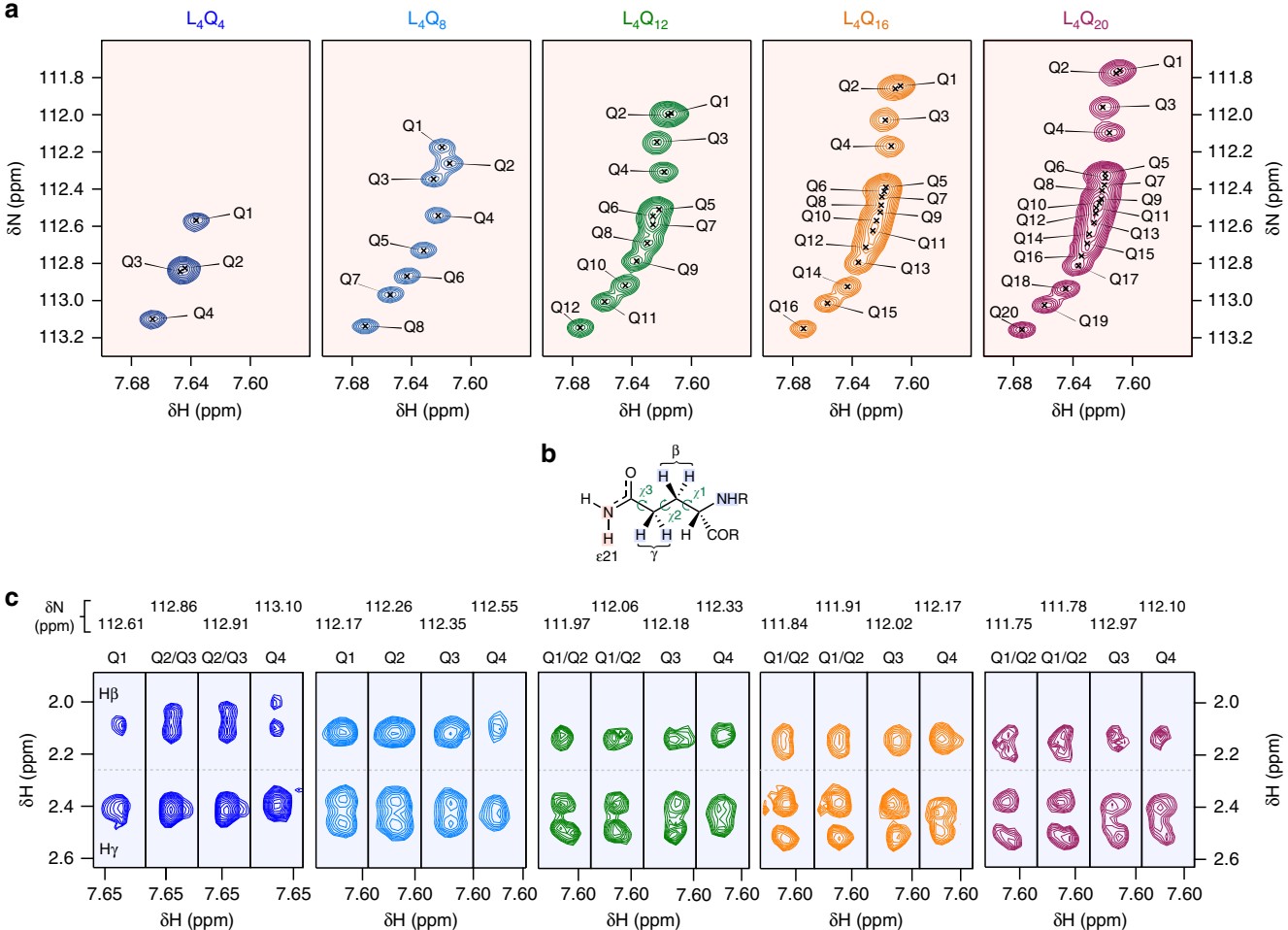

**Fig. 2** The conformations of the Gln side chains are well defined. **a** Expanded regions of the $^{1}$H,$^{15}$N heteronuclear single quantum correlation (HSQC) spectra of peptides $L_4Q_4$ to $L_4Q_{20}$ showing the H$\epsilon_{21}$ side chain resonances. **b** Structure of the Gln side chain with an indication of the nuclei whose resonances are shown in **a** (red shade) and in **c** (blue shade). **c** Regions of the $^{15}$N planes of the H(CC)(CO)NH spectra of peptides $L_4Q_4$ to $L_4Q_{20}$ measured at pH 7.4 and 278 K containing the side chain aliphatic $^{1}$H resonances of the first four residues (Q1–Q4) of the polyQ tract. All nuclear magnetic resonance (NMR) spectra were measured at pH 7.4 and 278 K

suggesting that it plays a role in stabilizing this secondary structure. In addition, for the reweighted MD ensembles of all the peptides, we observed that the frequency of this specific hydrogen bond decreases along the polyQ tract (Fig. 4b).

We analyzed the rotameric distributions of Gln residues involved in these hydrogen bonds in the reweighted trajectories and observed that they constrain the range of values of $\chi^1$ and $\chi^3$ that they can adopt (Fig. 4c). Note that while the distribution of $\chi^1$ in Gln residues in α-helices is generally bimodal[34], only $\chi^1$ values around −60° are compatible with the suggested H-bonding network, which also results in an enrichment of $\chi^3$ values around 90°. Remarkably, these observations agree with the NMR results, which point to the adoption of a specific conformation state by these side chains (Fig. 2c). Since the main chain chemical shifts used to reweight the MD trajectories do not directly report on side chain conformation, this result suggests that the main chain and side chain conformations are not independent, a property that is to some extent encoded in the MD force field. As an example, we show a frame of the trajectory obtained for peptide $L_4Q_{16}$ in which two such hydrogen bonds occur simultaneously (involving residues Q1 and Q4 but not Q2 and Q3; Fig. 4d). The NMR-derived structural ensembles thus suggest that sc(i) → mc(i − 4) hydrogen bonds can be part of a hydrogen bonding network in which the CO group accepts two hydrogen bonds donated by the Gln side (purple) and main (yellow) chains.

**Sc(i)→mc(i − 4) hydrogen bonds stabilize the polyQ helix**. To test the importance of sc(i) → mc(i − 4) hydrogen bonds, we used CD to analyze the secondary structure of a series of peptides based on the $L_4Q_{16}$ sequence with Gln residues mutated to Glu (Fig. 5a, b). Gln and Glu have similar structures and helical propensities[35], but the side chain of the latter is deprotonated at pH 7.4 and cannot act as a hydrogen bond donor. Decreases in helicity after mutation of Gln residues are thus compatible with the involvement of these residues in helix stabilization via sc(i) → mc(i − 4) hydrogen bonds. Since the frequency of such hydrogen bonds in the NMR-derived ensembles was highest at the N terminus of the tract (Fig. 4b), we analyzed the effect of mutating, one at a time, the first five Gln residues (peptides Q1E to Q5E). We found that the helicity of peptides Q1E to Q4E was lower than that of $L_4Q_{16}$. We observed a shift of the minimum at ca. 205–208 nm to lower wavelengths and a relative decrease in the ellipticity at 222 nm, which, together, accounted for a decrease in helicity from 40 to 30%. By contrast, we found that the helicity of Q5E was very similar to that of $L_4Q_{16}$ (Fig. 5c and Supplementary Fig. 11), thereby suggesting that the propensity of the first four Gln residues to donate a hydrogen is higher than that of the fifth one. This finding is in agreement with the $^{15}$N Gln side chain chemical shifts, where we observed especially low values for the first four residues, which could be attributed to particularly strong hydrogen bonding interactions (Fig. 2a). We also analyzed a

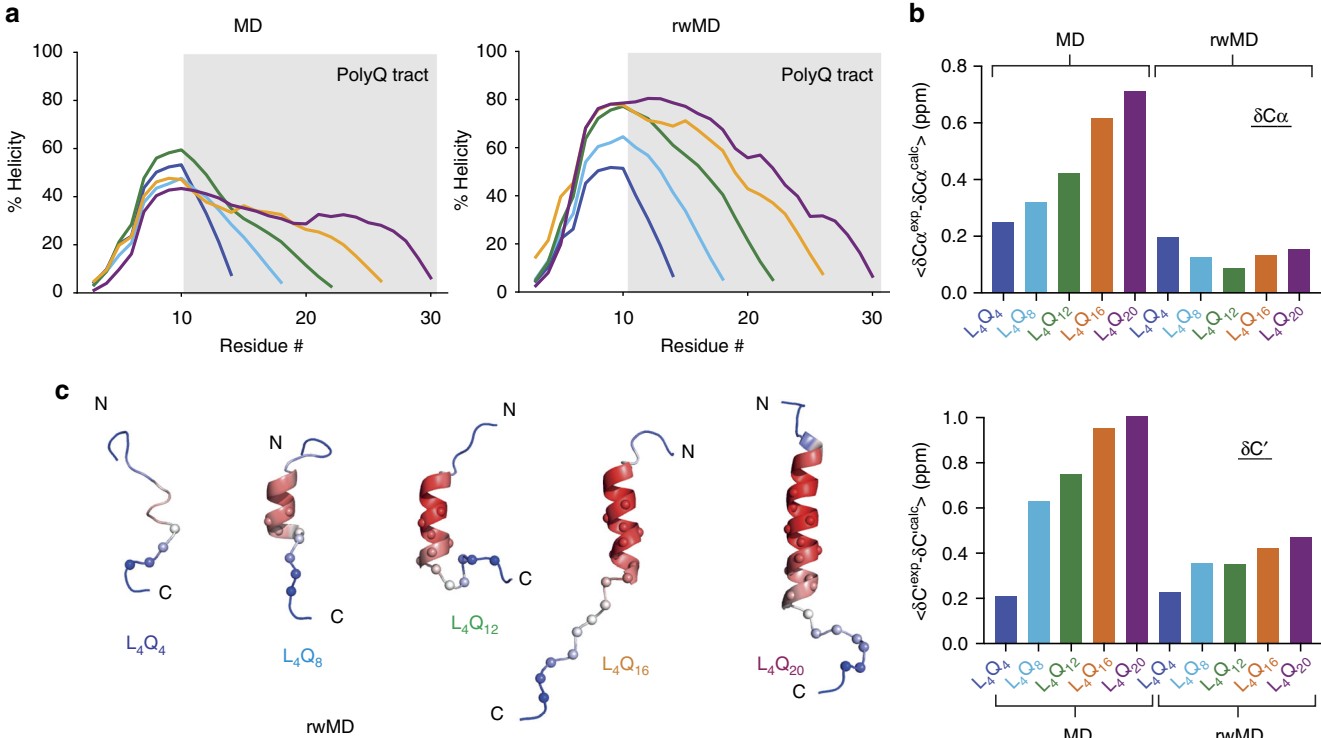

**Fig. 3** The structural properties of polyQ helices are tract length dependent. **a** Residue-specific helicity obtained for peptides $L_4Q_4$ to $L_4Q_{20}$ before (molecular dynamics (MD)) and after reweighting (rwMD) the trajectories obtained by MD on the basis of the backbone chemical shifts. **b** Comparison of the difference between the experimental and back-calculated Cα and C′ chemical shifts with those back-calculated from the reweighted MD trajectories obtained for peptides $L_4Q_4$ to $L_4Q_{20}$. **c** Representative structures for peptides $L_4Q_4$ to $L_4Q_{20}$, defined as the frame of each trajectory with residue-specific helicity most similar to the ensemble-averaged counterpart. Residues are colored as a function of their average helicity (in the reweighted ensemble) and the Cα atoms of Gln residues are shown as spheres

mutant where the first four hydrogen-bonded Gln residues were simultaneously mutated to Glu (Q1-4E) and found that in this case the loss of helicity was larger, from 40 to 20%, similar to the value found in $uQ_{25}$ (Fig. 5b, c and Supplementary Fig. 11).

Since the p$K_a$ of Glu side chains is ca. 4, decreasing the pH of solutions of peptide Q1-4E to 2 should lead to their protonation and re-establish their ability to form sc($i$) → mc($i-4$) hydrogen bonds. To test this hypothesis, we analyzed the secondary structure of peptides $L_4Q_{16}$ and Q1-4E at pH 2 and 277 K by CD. As expected, we observed no change in the secondary structure of $L_4Q_{16}$. In contrast, for Q1-4E, we observed that this peptide was strongly helical at low pH, more so than $L_4Q_{16}$ (Supplementary Fig. 12). This observation suggested that, when protonated, Glu side chains, due to their acidic character, have an even higher propensity than Gln residues to donate a hydrogen bond to the main chain CO of the residue at position $i-4$. These results validate our approach to investigate side chain to main hydrogen bonds by Gln to Glu mutations and, in addition, contribute to explaining the high helical propensity observed in host–guest experiments for protonated Glu residues, where they were found to be more helical than any other amino acid except Ala[35].

It is remarkable that the first side chains of the polyQ tract have a particularly high propensity to form sc($i$) → mc($i-4$) hydrogen bonds. The other Gln residues also form these bonds but to a lower extent as indicated, for example, by their side chain chemical shifts. One difference between these two sets of Gln residues is that the former are at position $i+4$ relative to Leu residues, whereas the latter are at position $i+4$ relative to other Gln residues (Fig. 5b). Since the strength of hydrogen bonds depends on their degree of shielding from water[36], we hypothesized that the sc($i$) → mc($i-4$) hydrogen bonds between

Gln and Leu residues are stronger, at least in part, due to shielding of water by Leu side chains. Indeed, as α-helices have 3.6 residues per turn, the sc($i$) → mc($i-4$) hydrogen bond between residues L1 and Q1 can be shielded by the side chain of residue $i$ (L1) (Fig. 5b). To study this point, we measured the helicity of a peptide based on the sequence of $L_4Q_{16}$ but with all Leu residues mutated to Ala (L1-4A), an amino acid that has a smaller side chain and, presumably, a lower ability to shield this hydrogen bond. Despite the higher intrinsic helical propensity of Ala compared to Leu[35] and the higher predicted helicity of L1-4A compared to $L_4Q_{16}$ (Supplementary Fig. 13), we found that substitution of the four Leu for Ala residues reduced the helical content from 40% in peptide $L_4Q_{16}$ to ca. 20% in L1-4E, a value as low as that of Q1-4E (Supplementary Fig. 14 and Fig. 5c). This finding confirms that the shielding properties of the Leu side chains are indeed key for the strength of this interaction and for its ability to stabilize polyQ helices. In addition, it indicates that accounting for the sc($i$) → mc($i-4$) hydrogen bond revealed in this work will be important to reliably predict the helicity of polyQ peptides on the basis of their sequences (Supplementary Fig. 13).

To confirm that the shielding provided by Leu is relevant for the ability of Gln to donate a hydrogen bond to the residue at relative position $i-4$, we characterized the synthetic peptide L1-4A by NMR. We compared the side chain $^1$H,$^{15}$N resonances of peptide L1-4A with those of $L_4Q_{16}$ by carrying out $^1$H,$^{15}$N-HSQC experiments at natural $^{15}$N abundance. We observed that there was a remarkable loss of dispersion in the $^{15}$N chemical shift dimension for L1-4A: except for the last three Gln residues, all other residues in the tract have the same $^{15}$N chemical shift (Fig. 5d). We then analyzed the aliphatic $^1$H resonances of the

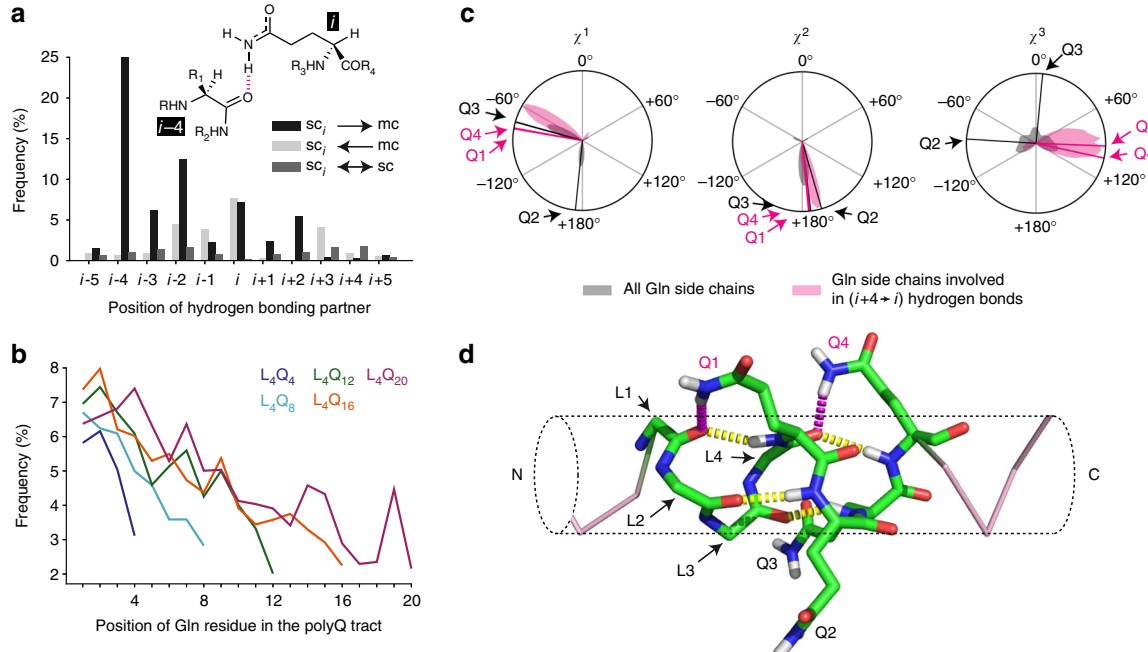

**Fig. 4** The structures of polyQ helices feature sc($i$) → mc($i$−4) hydrogen bonds. **a** Frequencies of the various types of hydrogen bonds involving Gln side chains in the aggregated reweighted trajectories. **b** Frequencies of sc($i$) → mc($i$ − 4) hydrogen bonds in the reweighted ensembles obtained for peptides $L_4Q_4$ to $L_4Q_{20}$ as a function of the position of the corresponding residue in the tract. **c** Distributions of the $\chi^1$, $\chi^2$, and $\chi^3$ dihedral angles of Gln side chains and of the subset of those side chains involved in sc($i$) → mc($i$−4) hydrogen bonds with an illustration of the values of the four side chains highlighted in **d**. **d** Frame of the trajectory obtained for $L_4Q_{16}$ where residues Q1 and Q4 (in purple), but not Q2 and Q3 (in black), are involved in sc($i$) → mc($i$ − 4) hydrogen bonds with the CO groups of residues L1 and L4, shown in purple, with an indication of the conventional mc($i$)→mc($i$ − 4) hydrogen bonds, shown in yellow. The Leu side chains are not shown, for clarity

Gln side chains and observed that, in contrast to $L_4Q_{16}$, the signals of Q1 to Q4 in L1-4A display collapsed γ and split β resonances, thus indicating that these side chains do not have the same conformation as in $L_4Q_{16}$.

**Backbone carbonyls form bifurcated hydrogen bonds**. Our results suggest that the side and main chains of Gln can simultaneously donate a hydrogen to the CO of the residue at relative position $i$ − 4 (Fig. 4d). This process can generate a type of bifurcate hydrogen bonding—shown to occur experimentally[37,38] and in QM calculations[38,39]—that takes advantage of the directionality of the lone pairs of the acceptor group. To model the sc($i$) → mc($i$ − 4) hydrogen bond accurately and measure their strength in terms of electron density, we performed MD simulations by means of the hybrid QM/MM methodology, which can in addition account for a series of effects that are overlooked in classical force fields such as lone pair directionality and electronic polarization. Specifically, given our results (Fig. 5b, c), the side chain carboxamide of the Gln residue at position $i$ and the main chain CO group of Leu at position $i$ − 4 in peptide $L_4Q_{16}$ were included in the QM subsystem, which was described at the DFT level of theory (see Fig. 6a). We performed a simulation of 150 ps at 300 K for $L_4Q_{16}$ started from a specific frame of the classical MD trajectory where the bifurcate bond is formed (Fig. 6), and focused our analysis on the interaction between Q1 and L1 (Fig. 5b).

Our analysis showed that the main chain to main chain hydrogen bond between Q1 and L1, mc(Q1) → mc(L1) is stable, that the sc(Q1) → mc(L1) bond can form reversibly, and that its breakage is caused by deviations of $\chi^3$ from the value required for the donor and acceptor to interact (+90 ± 30°, Figs. 4c, d and 6b, c). To study how the sc(Q1) → mc(L1) bond affects the mc(Q1) → mc(L1) interaction, we compared the effect of the former on the distribution of donor to acceptor distances in the latter.

The unconventional side chain to main chain bond causes the distribution to shift to longer distances, by 0.17 Å, thus weakening the main chain to main chain bond. This observation thus indicates that the main chain and side chain NH groups of Q1 compete for the main chain CO group of L1 (Supplementary Fig. 15). We then evaluated the strength of these interactions in terms of electron density at the natural bond critical point of the interaction, $\rho(r)$[40,41]. In the absence of the sc(Q1) → mc(L1) bond, the mc(Q1) → mc(L1) bond has an average density of 0.012 a.u. and, in its presence, of 0.008 a.u. By contrast, even in the presence of the mc(Q1) → mc(L1) bond, the value for the sc(Q1) → mc(L1) interaction is instead 0.017 a.u. on average, in agreement with the notion that the Gln side chain is a better donor than the main chain[32]. Importantly, the total density of the bifurcate hydrogen bond is on average 0.025 a.u. (Fig. 6c), thereby indicating that the cumulative interaction between Q1 and L1 is strong. These results show that the unconventional sc($i$) → mc($i$ − 4) hydrogen bonding interactions revealed in this work are bifurcate with the conventional mc($i$) → mc($i$ − 4) interactions and enhance the stability of polyQ helices.

## Discussion

Our results indicate that unconventional sc($i$) → mc($i$ − 4) hydrogen bonds donated by Gln side chains can stabilize the α-helices formed by polyQ tracts. Furthermore, we have also found that the strength of these bonds is determined by the residue type of the acceptor: Leu residues are good acceptors while Ala residues are not. These results help explaining the residue-specific structural properties of polyQ tracts reported in the recent literature[20,42,43]. We found that the four Leu residues flanking the polyQ tract of the AR at its N terminus are key for helicity[20] and can now attribute this to their high propensity to accept sc($i$) → mc($i$ − 4) hydrogen bonds. The tract of huntingtin also displays

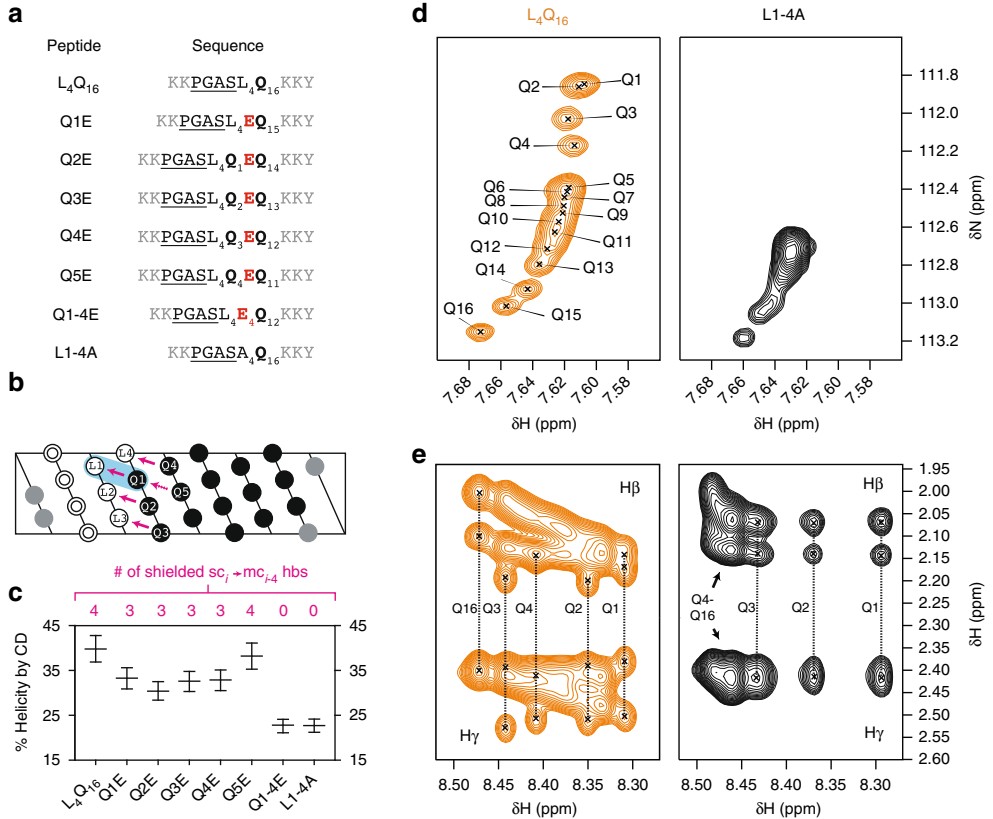

**Fig. 5** The sc($i$) → mc($i$ - 4) hydrogen bonds stabilize polyQ helices. **a** Peptides used to determine the contribution of sc($i$) → mc($i$ − 4) hydrogen bonds to the stability of helical secondary structures and the effect of shielding. **b** Projection of the fully helical structure of peptide $L_4Q_{16}$ in the color code used in **a**, where Lys residue areas are shown in gray, double circles represent the PGAS N-capping motif, Leu residues are displayed in white, and Gln residues in black, with an indication of the sc($i$) → mc($i$ − 4) hydrogen bonds in purple and of the shielding of the hydrogen bond between residues Q1 and L1 by the side chain of L1, as a blue shade. One unshielded, weaker hydrogen bond between Q1 and Q5 is represented by a dashed purple arrow. **c** Helicity of the peptides listed in **a** as determined by circular dichroism (CD) at pH 7.4 and 277 K, with an indication of the number of shielded hydrogen bonds that can occur in each peptide according to the model shown in **b**. The vertical bars represent the values of helicity obtained after scaling the experimental spectra by factors 0.9 and 1.1 to account for experimental error in the determination of peptide concentration. **d** Regions of the $^1$H,$^{15}$N heteronuclear single quantum correlation (HSQC) spectrum, measured at pH 7.4 and 278 K, of peptides $L_4Q_{16}$ (see also Fig. 2b) and L1-4A containing the H$\epsilon_{21}$ side chain resonances. **e** Region of the total correlation spectroscopy (TOCSY) spectrum of peptides $L_4Q_{16}$ and L1-4A measured at pH 7.4 and 278 K illustrating the β and γ $^1$H resonances of the Gln side chains

some helicity at low pH[42,43], although lower than that observed in the AR. Even though the ability of each particular type of residue to act as a sc($i$) → mc($i$ − 4) hydrogen bond acceptor remains to be determined, the observation that only the first position in the four-residue stretch—defining a turn of an α-helix—preceding the polyQ tract in huntingtin is a Leu residue could explain the lower secondary structure content of this tract.

The helical character of the polyQ tract is not homogeneously distributed in either the AR or huntingtin and is instead found to decrease gradually from the N terminus to the C terminus of the tract[20,42,43]. Our results indicate that this can be explained by a low propensity of Gln residues, relative to that of residues flanking the tracts at their N terminus, to accept sc($i$) → mc($i$ − 4) hydrogen bonds. Unless interrupted by residues with a high propensity to accept such bonds, such as Leu, helicity will decay towards the C terminus of the tract. In addition, our results provide a mechanistic interpretation of the results obtained by Kandel, Hendrickson and co-workers[2] regarding the effect of increasing the coiled coil character of polyQ tracts by interrupting them with Leu residues. These authors found that the peptides were fully helical and remained so after dissociation of the coiled coil upon heating to temperatures as high as 348 K due, we propose, to the presence of sc($i$) → mc($i$ − 4) hydrogen bonds, with Leu acting as acceptor.

We attribute the high propensity of Leu residues to accept sc($i$) → mc($i$ − 4) hydrogen bonds to the proximity of the hydrogen bond to the Leu side chain. This can prevent water molecules from donating a hydrogen to the carbonyl of that residue and, due to high energetic cost associated with the presence of unpaired hydrogen bonding partners[36], greatly strengthen its interaction with the Gln side chain at position $i$ + 4 despite the entropic cost of constraining its conformation (Figs. 4 and 6). Relatively dry environments where phenomena equivalent to this can occur include the core of globular proteins[44] and the interior of cell membranes[45], as well as amyloid fibrils, where hydrogen bonding interactions involving Gln and Asn side chains parallel to the fibril axis contribute to the stability of the quaternary structure[46]. In addition, it has been shown that both exon 1 of huntingtin[47] and the transactivation domain of the AR[48] form condensates that define environments of low dielectric constant, where electrostatic interactions may be strongly favored[49]. It will be interesting to address whether this phenomenon plays a role in the highly cooperative liquid–liquid phase separation process of these and similar proteins, as has been proposed to be the case for protein folding, where considering the context-dependent strength of inter-residue interactions proved important for reproducing the high cooperativity of protein folding in lattice simulations[50].

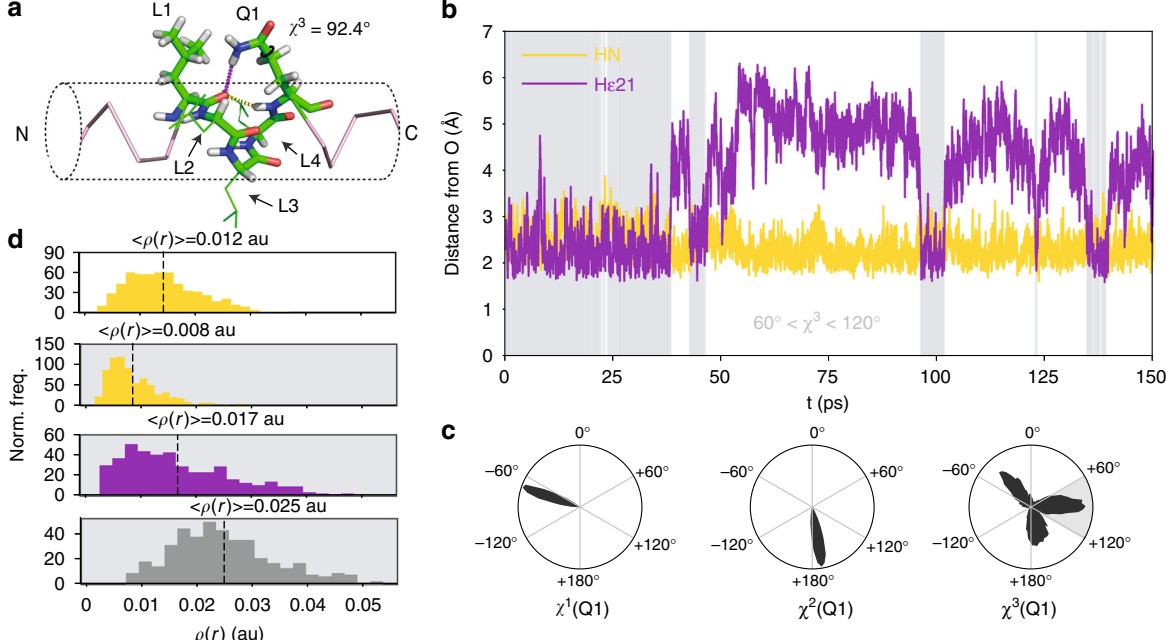

**Fig. 6** The sc($i$) → mc($i-4$) and mc($i$) → mc($i-4$) hydrogen bonds are bifurcate. **a** Starting configuration used in the simulation, with the atoms included in the QM subsystem shown as sticks and the distances plot in **b** shown as dashed lines. **b** Time series of the distances between donor and acceptor for the mc(Q1) → mc(L1) and sc(Q1) → mc(L1) interactions, with an indication, in a gray background, of the frames for which $60° < \chi^3 < 120°$. **c** Distributions of the $\chi^1, \chi^2$, and $\chi^3$ dihedral angles of the side chains of Q1 with an indication, as a gray shade, of the range of values of $\chi^3$ that are compatible with the sc(Q1) → mc(L1) hydrogen bond. **d** Distribution, plotted as a normalized histogram, of the electron density $\rho(r)$ corresponding to the mc(Q1) → mc(L1) interaction (yellow) in the absence (white background) and in the presence (gray background) of the sc(Q1) → mc(L1) interaction and of the electron density corresponding to the sc(Q1) → mc(L1) interaction (purple) and bifurcate interactions (gray)

PolyQ tracts are frequently found in transcriptional regulators, particularly in transcription factors[1]. In several cases, the transcriptional activity of these molecules has been found to be dependent on the length of the polyQ tracts that they harbor; the physical basis of this phenomenon has, however, not been firmly established to date[1,51]. Our results provide a plausible rationale as they suggest that variations in the length of polyQ tracts result in changes in the secondary structure of the transactivation domain of transcription factors. Indeed, such variations can affect the strength of the protein–protein interactions[52], particularly of those regulating transcription[53], which include interactions with transcriptional co-regulators and with general transcription factors. Whether a certain change in tract length causes a decrease or an increase in activity might depend on whether the polyQ tract and its flanking regions are involved in interactions with transcriptional co-activators or co-repressors and should therefore be context-dependent, as found experimentally[51,54].

A number of in vitro experiments have established that the formation of fibrillar aggregates by proteins bearing polyQ tracts can proceed via oligomers[55], potentially liquid-like[56], stabilized by intermolecular interactions between flanking regions of polyQ tracts and equivalent to those stabilizing coiled coils[2,57,58]. In proteins bearing polyQ tracts such as huntingtin[59,60] and AR[20,21] extending the length of the tract increases its helicity and, as we have now shown, that of the sequence immediately flanking them at the N terminus. Remarkably, this also appears to be the case when they are studied in the context provided by the domains where they are found[20,21] and, indeed, in that provided by full-length protein[60]. It is therefore conceivable that this extension will change the secondary structure and thus the strength of the interactions that stabilize these oligomers[52], and, potentially, the rate at which they convert into fibrils. Our data thus suggest that tract expansion can alter the structure and the stability of the

oligomers populated on the fibrillization pathway and consequently modify the rate at which toxic fibrillar species build up[14].

In summary, we have shown that side chain to main chain hydrogen bonds donated by Gln side chains can substantially increase the helical propensity of polyQ sequences. In addition, we found that, for a given sequence context, tract expansion increases helical propensity due to the accumulation of these unconventional interactions. Such hydrogen bonds, that are due to high propensity of the carboxamide group of the Gln side chain to donate hydrogens, are so energetically favored that they can offset the entropic cost of constraining the range of conformations available to the side chain. In addition, we have shown that the strength of these interactions depends on the degree to which the Gln side chains are exposed to water, implying that the secondary structure of polyQ tracts may vary depending on solution conditions, oligomerization state, and interactions with other molecules. Our findings provide a plausible mechanistic explanation for the link between polyQ tract length, AR transcriptional activity and solubility, and for the range of AR polyQ tract lengths found in the healthy male population. More generally, they suggest that changes in helicity may underlie the effect of tract length changes on transcriptional activity and on aggregation via helical oligomeric intermediates in polyQ diseases.

## Methods

**CD experiments**. All synthetic peptides were obtained as lyophilized powders with >95% purity from GenScript (Piscataway, NJ, USA) with free N and C termini. They were dissolved in 6 M guanidine thiocyanate (Merck KGaA, Darmstadt, Germany) and incubated overnight at 298 K to ensure that the resulting solutions were monomeric. The denaturant was removed by size exclusion chromatography in an Äkta Purifier system (GE Healthcare, Chicago, IL, USA) equipped with a Superdex Peptide 10/300 GL column equilibrated in MilliQ water with 0.1% trifluoroacetic acid. The fractions corresponding to the monomeric peptides were collected, pooled, and centrifuged at 104,000 rpm for 3 h in an Optima TLX

tabletop ultracentrifuge equipped with a TLA 120.1 rotor (Beckman Coulter, Atlanta, GA, USA). Sodium phosphate buffer was added to a final concentration of 20 mM, and the samples were adjusted to pH 7.4 prior to quantification and analysis by CD. The former was performed by reversed-phase chromatography in an Agilent 1200 HPLC system (Agilent Technologies, Santa Clara, CA, USA) equipped with a Phenomenex Jupiter 5 μm C18 300 Å column (Torrance, CA, USA) or, for the peptides with a Tyr residue, by measuring the absorbance at 280 nm by using 1490 $cm^{-1} M^{-1}$ as value of the Tyr molar extinction coefficient. CD spectra were acquired on 30 μM samples in a Jasco 815 UV spectro-photopolarimeter at 277 K with a 1 mm optical path cuvette, and their deconvolution to determine secondary structure propensities was performed with the analysis programme CONTIN (reference set 7) hosted at DichroWeb[61] (dichroweb.cryst.bbk.ac.uk). In Fig. 5d, to estimate the uncertainty in the helicity values obtained in this deconvolution, which relies on an accurate quantification of the peptide concentration, in addition to the value obtained without scaling the experimental spectrum, we plot those obtained after scaling it by factors 0.9 and 1.1.

**NMR experiments**. Synthetic genes coding for peptides $L_4Q_4$ to $L_4Q_{20}$ (Fig. 1a) fused to His$_6$-SUMO and codon-optimized for expression in *Escherichia coli* (Supplementary Table 1) were obtained cloned in a pDEST-17 expression vector from GeneArt (Thermo Fisher Scientific, Waltham, MA, USA). The corresponding constructs were expressed in Rosetta (DE3)pLysS *E. coli*-competent cells—Novagen (Merck), in M9 medium containing $^{15}NH_4Cl$ and $^{13}C$-glucose as sole nitrogen and carbon sources, obtained from Cambridge Isotope Laboratories, Inc. (Tewksbury, MA, USA). After cell lysis in a 20 mM Tris-HCl buffer containing 100 mM NaCl and 20 mM imidazole, the soluble fractions were purified by immobilized metal affinity chromatography (IMAC) in an Äkta Purifier System (GE Healthcare, Chicago, IL, USA) equipped with a HisTrap HP 5 mL column. The eluted fractions containing the His$_6$-SUMO-tagged peptides were pooled and dialyzed against the lysis buffer to remove imidazole before digesting them with SUMO protease (0.05 mg/mL). Cleaved peptides were further purified by a second IMAC step and dialyzed against pure MilliQ water before lyophilization. The lyophilized recombinant $^{15}N$-$^{13}C$-enriched peptides were treated in the same way as the synthetic ones to prepare 100 μM samples for the NMR experiments, which were in all cases carried out in a 600 MHz Bruker Avance spectrometer equipped with a cryoprobe. The samples contained 10 μM DSS (4,4-dimethyl-4-silapentane-1-sulfonic acid) for chemical shift referencing. The backbone resonances of peptides $L_4Q_4$ to $L_4Q_{20}$ were assigned using three-dimensional (3D) triple resonance experiments (HNCO, HN(CA)CO, HN(CO)CA, HN(CO)CACB) acquired with NUS at 278 K. The side chain resonances were assigned with 3D H(CC)(CO)NH, (H)CC(CO)NH experiments. NMR experimental data were processed using qMDD[62] for non-uniform sampled data and NMRPipe[63] for all uniformly acquired experiments. The synthetic peptides $L_4Q_{16}$ and L1-4A were prepared as detailed above to reach a final concentration of 250 μM and characterized by two-dimensional homonuclear (total correlation spectroscopy (TOCSY) and nuclear Overhauser effect spectroscopy (NOESY)) and heteronuclear ($^1H,^{15}N$-HSQC, at natural $^{15}N$ abundance) experiments. The TOCSY and NOESY mixing times were 70 and 200 ms, respectively.

**Molecular dynamics**. Input coordinates were generated using MacPyMOL in fully helical conformations. All simulations were performed in MD simulation software ACEMD[64] using the CHARMM22*[65] force field, which was designed to have an accurate helix-coil balance. Each system was explicitly solvated in the TIP3P water model inside cubic boxes from 25 to 40 Å distance around the peptides, depending on their length, and neutralized with Cl$^-$ and Na$^+$ ions. Initial conformations were minimized and equilibrated under NPT conditions at 1 atm and 300 K for 1 ns. Production simulations were performed at 300 K in the NVT ensemble using a 4-fs time-step for up to 5 μs.

**Analysis and trajectory reweighting by maximum entropy**. The secondary structure of individual frames was analyzed with DSSP[66], and the chemical shifts were back-calculated with the predictor PPM[67]. The trajectories were reweighted to match the experimental chemical shifts by means of a BME method[30]. The BME approach contains a single, free parameter ($\theta$) that determines the balance between fitting the experimental data and not deviating too much from the prior information encoded in the force field. We chose $\theta = 4$ for the analysis shown in the main text, as an analysis revealed that this value provides a good balance between the two terms (Supplementary Fig. 9). The results for other values of $\theta$ are shown in Supplementary Fig. 8.

**QM/MM calculations**. The starting structure was selected from the classical MD simulations of $L_4Q_{16}$, preserving the previously defined box of water and ions. The AMBER 16 program[68] interfaced to the Terachem 1.9 program (www.petachem.com, accessed June 1, 2017) was used for the QM/MM simulation. QM atoms were described at the BLYP/6-31G* level, including a dispersion correction[69]. The classical subsystem was described with the CHARMM22*[65] force field by making use of the Chamber keyword of Parmed program included in AMBERTOOLS 16[68]. The link atoms procedure, as implemented in AMBER program, was used to saturate the valence of the frontier atoms. Periodic boundary conditions were

employed with an electrostatic cutoff of 12 Å. A time-step of 1 fs was employed. The structure was first minimized and then equilibrated for 10 ps in a QM/MM-MD run. A production run was then performed with a total simulation time of 150 ps. The Natural Bond Critical Point analysis[70,71] was performed with the NBO 6.0 program[72].

**Hydrogen bond criteria**. To classify whether two atoms are hydrogen bonded, we used angle and distance criteria. Specifically, we defined hydrogen bonds as those where the distance between the donor and the acceptor was shorter than 3.4 Å (2.4 Å between H and heavy atom) and the donor hydrogen–acceptor angle was >120°.

**Model structures**. After reweighting, we calculated the residue-specific helicity for all of the peptides using the algorithm DSSP[66]. For model structure selection, residues that were in the helical conformation in more than 50% of the simulation were defined as helical and the rest as random coil. From the simulation, the structures that fit this definition were selected and colored on the basis of their average helicity, shown in Fig. 3c. Color scale goes from dark blue (0% helicity) to dark red (78% helicity).

**Reporting summary**. Further information on research design is available in the Nature Research Reporting Summary linked to this article.

## Data availability
The chemical shifts of peptides $L_4Q_4$, $L_4Q_8$, $L_4Q_{12}$, $L_4Q_{16}$, and $L_4Q_{20}$ have been deposited in the BMRB (www.bmrb.wisc.edu) with accession codes 27713, 27714, 27715, 27716, and 27717. Other data are available from the corresponding authors upon reasonable request.

## Code availability
The code used to reweight the ensembles is available at https://github.com/KULL-Centre/BME

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

## Acknowledgements

We thank Sandro Bottaro, Ernest Giralt, Gerhard Hummer, Víctor Muñoz, and Huan-Xiang Zhou for helpful discussions and the ICTS NMR facility, managed by the scientific and technological centers of the University of Barcelona (CCiT UB), for their help in NMR. K.L.-L. and M.B.A.K. acknowledge funding from the Lundbeck Foundation and the BRAINSTRUC initiative. B.T. and J.A. acknowledge, respectively, FPI and Juan de la Cierva fellowships from MINECO. R.P and I.C.F. acknowledge funding from the European Commission (iNEXT, 653706). M.O. acknowledges the Spanish Ministry of Science (BFU2014-61670-EXP), the Catalan SGR, the Instituto Nacional de Bioinformática, the Biomolecular and Bioinformatics Resources Platform (ISCIII PT 13/0001/0030) co-funded by the Fondo Europeo de Desarrollo Regional (FEDER). R.C. acknowledges funding from MINECO (CTQ2016-78636-P). X.S. acknowledges funding from AGAUR (2017 SGR 324), Marató TV3 (102030), MINECO (BIO2012-31043 and BIO2015-70092-R), and the European Research Council (CONCERT, contract number 648201). IRB Barcelona is the recipient of a Severo Ochoa Award of Excellence from MINECO (Government of Spain).

## Author contributions

A.E., B.T., J.G., J.A., G.C., D.M. and X.S. performed experiments and simulations, and analyzed and interpreted the results. M.B.A.K., G.B.-S., B.E., M.G., R.P., I.C.F., T.D., O.M., M.O. and R.C. contributed to performing, analyzing, and interpreting the results. M.B.A.K., K.L.-L., J.A. and M.O. contributed tools. A.E., B.T., J.G., R.C., K.L.-L. and X.S. established the hypothesis, designed the experiments, and led their analysis and interpretation. X.S. conceived and led the project and wrote the first draft of the manuscript. All authors contributed to the final version.

## Additional information

**Competing interests:** The authors declare no competing interests.

