## [Peer Review File · Nature Communications]

REVIEWERS' COMMENTS:

Reviewer #2 (Remarks to the Author):

The authors have addressed the concerns expressed in my previous referee report quite adequately. However, there is an error in the added ref.50 that must be corrected before publication. This reference should be to the Phys Rev Lett (2000) paper of Kaya and Chan, because the Proteins (2000) paper did not address context-dependent interactions. It's the Phys Rev Lett (2000) paper that contains an explicit model of context-dependent interactions (see Fig.2 of the Phys Rev Lett ref below), as stated clearly in my previous report. It's puzzling why this error appeared. The correct reference for ref.50 is:

Kaya, H. & Chan, H. S. Energetic components of cooperative protein folding. Phys. Rev. Lett. 85, 4823-4826 (2000).

Once this correction has been made, this revised manuscript should be published.

Referee 2

The authors have addressed the concerns expressed in my previous referee report quite adequately. However, there is an error in the added ref.50 that must be corrected before publication. This reference should be to the Phys Rev Lett (2000) paper of Kaya and Chan, because the Proteins (2000) paper did not address context-dependent interactions. It's the Phys Rev Lett (2000) paper that contains an explicit model of context-dependent interactions (see Fig.2 of the Phys Rev Lett ref below), as stated clearly in my previous report. It's puzzling why this error appeared. The correct reference for ref.50 is: Kaya, H. & Chan, H. S. Energetic components of cooperative protein folding. Phys. Rev. Lett. 85, 4823-4826 (2000). Once this correction has been made, this revised manuscript should be published.

We were glad to read that Reviewer 2 found that we had addressed her/his concerns quite adequately and that in her/his opinion the revised version should be published in *Nature Communications*. In addition we thank the Reviewer for pointing out that we referred to the wrong 2000 paper: this was due to an error in using our reference management system that we have corrected in the final version.